# First and second wave dynamics of emergency department utilization during the COVID-19 pandemic: A retrospective study in 3 hospitals in The Netherlands

Robi Dijk[1⊙], Patricia Plaum[2⊙], Stan Tummers[3], Frits H. M. van Osch[4,5]*, Dennis G. Barten[1], Gideon H. P. Latten[3]

1 Emergency Department, Viecuri Medical Centre, Venlo, The Netherlands, 2 Intensive Care Unit, Viecuri Medical Centre, Venlo, The Netherlands, 3 Emergency Department, Zuyderland Medical Centre, Heerlen, The Netherlands, 4 School of Nutrition and Metabolism (NUTRIM), Maastricht University, Maastricht, The Netherlands, 5 Department of Clinical Epidemiology, Viecuri Medical Centre, Venlo, The Netherlands

⊙ These authors contributed equally to this work.
* f.vanosch@maastrichtuniversity.nl

**Data Availability Statement:** All relevant data are within the paper and its Supporting Information files.

## Abstract

### Objective

During certain phases of the COVID-19 pandemic, a decrease was observed in emergency department (ED) utilization. Although this phenomenon has been thoroughly characterized for the first wave (FW), second wave (SW) studies are limited. We examined the changes in ED utilization between the FW and SW, compared to 2019 reference periods.

### Study design and methods

We performed a retrospective analysis of ED utilization in 3 Dutch hospitals in 2020. The FW and SW (March-June and September–December, respectively) were compared to the reference periods in 2019. ED visits were labeled as (non-)COVID-suspected.

### Results

During the FW and SW ED visits decreased by 20.3% and 15.3%, respectively, when compared to reference periods in 2019. During both waves high urgency visits significantly increased with 3.1% and 2.1%, and admission rates (ARs) increased with 5.0% and 10.4%. Trauma related visits decreased by 5.2% and 3.4%. During the SW we observed less COVID-related visits compared to the FW (4,407 vs 3,102 patients). COVID-related visits were significantly more often in higher need of urgent care and ARs were at least 24.0% higher compared to non-COVID visits.

### Conclusion

During both COVID-19 waves, ED visits were significantly reduced. ED patients were more often triaged as high urgent, the ED length of stay was longer and ARs were increased compared to the reference period in 2019, reflecting a high burden on ED resources. During the

**Funding:** The VieCuri Corona Foundation and Regio Noord-Limburg have supported this study by funding part of the Open Access cost associated with publication.

**Competing interests:** The authors have declared that no competing interests exist.

**Abbreviations:** AR, admission rate; COVID-19, coronavirus disease; ED, emergency department; FW, first wave; GP, general practitioner; IQR, interquartile Range; MTS, Manchester Triage System; PPE, personal protective equipment; SARS-CoV-2, severe acute respiratory syndrome coronavirus 2; SW, Second wave; WHO, World Health Organization.

FW, the reduction in ED visits was most pronounced. Here, ARs were also higher and patient were more often triaged as high urgency. These findings stress the need to gain better insight into the motives of patients to delay or avoid emergency care during pandemics, as well as to better prepare EDs for future outbreaks.

## Introduction

Since the first cases of SARS-CoV-2 emerged in Wuhan in November 2019, the resultant coronavirus disease (COVID-19) has spread rapidly around the world. In March 2020, the World Health Organization (WHO) declared the outbreak a pandemic [1]. In response to the steep increase of COVID-19 cases in 2020, hospital organizations prepared for increasing patient volumes by changing workforce and infrastructure with particular focus on emergency departments (EDs) [2].

In the Netherlands, the first COVID-19 diagnosis was confirmed on February 27[th], 2020. Government regulations were increased stepwise until a partial lockdown was imposed on March 23[rd], 2020. This included the closure of public institutions (such as schools) and stay-at-home policies for those with possible COVID-19 infection.

At the end of the first wave (FW, March to June 2020) a total of 50,273 Dutch citizens had tested positive for COVID-19. However, the true number of infections probably was significantly higher because of restricted testing policies, caused by limited testing capability during the FW. 11,397 patients were admitted to a hospital (of which 2,939 (25.8%) to an intensive care unit (ICU)), and 10,067 people died due to COVID-19 [3–5].

Most restrictions were lifted after a rapid drop in COVID-19 infections and hospital admissions in July and August 2020. By September, however, infection rates started to spike again. As a consequence, a second lockdown was enforced [6]. During the second wave (SW, September to December 2020) a total of 726,314 had a confirmed SARS-CoV-2 infection. In the SW, 19,354 patients were admitted to a hospital (3,629 (18.8%) to an ICU), and 10,046 people died from a COVID-19 infection [3, 4].

In the early stages of the pandemic, citizens were advised to only seek care when absolutely necessary [7]. Concerns about non-COVID-19 emergencies were subsequently raised when a 30% reduction in ED visits was observed [8, 9]. It was hypothesized that this reduction was due to both COVID-19 fears and lockdown effects, including reduced exposure to injury-prone activities, improved hygiene measures in the community and downscaling of non-acute healthcare [10]. Other studies found that a relatively large proportion of ED patients reported delay in seeking emergency care [11, 12]. During the SW, elective-non-COVID-care was continued as long as hospital capacity allowed for. In contrast to the FW, studies on ED utilization during the SW are limited.

As the pandemic continues and subsequent waves arise, it is of utter importance to gain better insight into the causes and consequences of healthcare utilization patterns during this public health crisis. Only then we can deliver appropriate care and minimize loss of health due to avoiding urgent care consultations.

In this retrospective observational study, we investigated ED utilization and patient volumes during the FW and SW of the COVID-19 pandemic in the Netherlands, and examined differences between those waves and pre-COVID reference periods.

## Methods

### Design and setting

In this retrospective observational study, we investigated the utilization of 3 hospital-based EDs in the southeast of the Netherlands during the first and second wave of the COVID-19

pandemic (from March-June and September–December 2020, respectively). Identical periods in 2019 were used as a reference. The 3 EDs combined serve a population of 760,000 individuals. Annual census is 35,000 patients for ED 1, 25,000 for ED 2 and 25,000 for ED 3.

## Patients

All patients who visited one of the EDs during the study period were eligible for inclusion.

Patients were excluded when it was unclear whether the ED visit was COVID-19 related and when patients refused participation in retrospective studies. In addition, we excluded patients in which ED length of stay was likely to be documented incorrectly, either 0 minutes or longer than 10 hours were considered to be outliers.

If patients previously declared they do not want to participate in any studies, it is documented in their patient records. These patient records were filtered out before we received the data. For all other patients, it was not reasonable to obtain informed consent due to the number of patients. Also, patients received standard treatment and they did not experience any benefits, harms or risks from this study. The ethics committee waived the need for informed consent.

Data was automatically extracted from the electronic health record from patients and only re-identifiable by the researchers.

## Analysis and statistics

We retrieved information from standard digital patient records: age, gender, date of ED visit, possible COVID-19 infection, triage urgency (using the Manchester Triage System, MTS) [13], ED length of stay, referral route, admission and whether the ED visit was trauma related or not. Patients triaged red or orange using the MTS are classified as high urgency visits (seen by doctor within 10 minutes). Yellow, green and blue triage colours (seen by doctor within 1–3 hours) are classified as low urgency visits. From April-May 2020, the EDs prospectively labelled patients as COVID-19 suspected using the criteria in Table 1. We retrospectively retrieved the missing labels for possible COVID-19 infection in patients who presented before hospitals had implemented prospective tracking, using identical criteria (Table 1).

We compared the ED utilization between the FW and SW and between 2020 and their respective reference periods in 2019.

Data were analyzed using IBM Corp. Released 2020. IBM SPSS Statistics for Macintosh, Version 27.0. Armonk, NY: IBM Corp. Descriptive analyses were used for patient characteristics. Continuous data were reported as means with standard deviation (SD) and compared

**Table 1. Criteria for possible COVID-19 infection.**

| Criteria |
| --- |
| Fever |
| Dyspnoea |
| Cough or respiratory complaints |
| Contact with positive patient |
| Chest pain |
| Unexplained nausea, vomiting or abdominal pain |
| Positive test or awaiting test result |
| Unexplained diarrhea |
| Sensory impact of taste and/or smell |
| Patients whereby history taking was not possible or reliable (e.g. cardiac arrest, unconscious or delirium) |

using Students' T test, or as medians with interquartile ranges (IQR) and compared using the Mann Whitney U test. Categorial data was reported as absolute numbers and as valid percentages (to correct for missing data); they were compared using chi-square or Fisher exact tests. Since multiple comparisons were made in the analysis, after Bonferroni correction a p-value of 0.05/number of comparisons was considered statistically significant.

The study was approved by the medical ethical committee of Zuyderland Medical Center, Heerlen, the Netherlands (METCZ20210031).

## Results

In total, 127,060 patients who visited any of the three EDs in 2019 and 2020 were eligible for inclusion; 4,403 (3.5%) were considered outliers and were excluded (4,362 0-minute visits and 41 more than 10 hours).

Of these eligible patients, 56,719 visited one of the three EDs in the year 2020, compared to 65,938 patients in 2019. Of all included patients, the median age was 60.5 years (IQR 34–76) and 50.7% was male. In total, 9,034 (16.0%) of the ED patients in 2020 were labelled as possible COVID-19 infected, of which 4,777 presented during the FW and 3,102 during the SW (Fig 1).

### First wave compared to second wave

During the FW, 18,024 patients presented to the ED compared to 18,282 during the second wave. Compared to 2019, there was a decline of 20,3% during the first wave and a decline of 15.2% during the second wave as shown in Fig 1. When comparing the FW and SW, the number of patients with possible COVID-19 infections was higher during the FW (4,777, 26.5%) compared to the SW (3,102, 17.0%, p<0.001). Furthermore, an increase in trauma related visits (27.4% vs 28.7%, p<0.001) and a decrease in admission rates (AR) during the SW was observed (57.3% vs 56.7%, p<0.001).

### Comparison between 2020 and 2019

The dynamics of ED utilization during the two waves and their respective reference periods in 2019 are shown in Table 2. Patients transported to the ED by ambulance increased by 6.6% (p<0.001) and 5.0% (p<0.001) in the FW and SW, respectively. Furthermore, the percentage of patients triaged as high urgency increased by 3.1% in the FW (p<0.001) and 2.1% in the SW (p<0.001). Trauma related visits decreased by 5.2% in the FW (p<0.001) During the SW we observed a decrease of 3.4% in trauma related visits, but this was not statistically significant after Bonferroni correction (p = 0.004).

Median ED length of stay was longer during both waves (P<0.001) compared to 2019, yet the median increase in length of stay was 3 minutes longer during the FW compared to the SW (P<0.001) Finally, AR increased by 5.0% (p<0.001) and 10.4% (p<0.001) during the first and second pandemic wave, respectively.

### Possible COVID-19 versus non-COVID-19

Table 3 depicts a comparison between possible COVID-19 vs non-COVID-19 visits during the FW and SW. High urgent visits were 18.2% (p<0.001) and 26.4% (p<0.001) more common in patients with possible COVID-19 compared to non-COVID-19 visits during the first and second wave, respectively.

During the first and second wave, the AR for suspected COVID-19 patients was significantly higher at 74.9% and 79.9% compared to 50.1% (p<0.001) and 51.5% (p<0.001) for non-COVID-19 suspected patients.

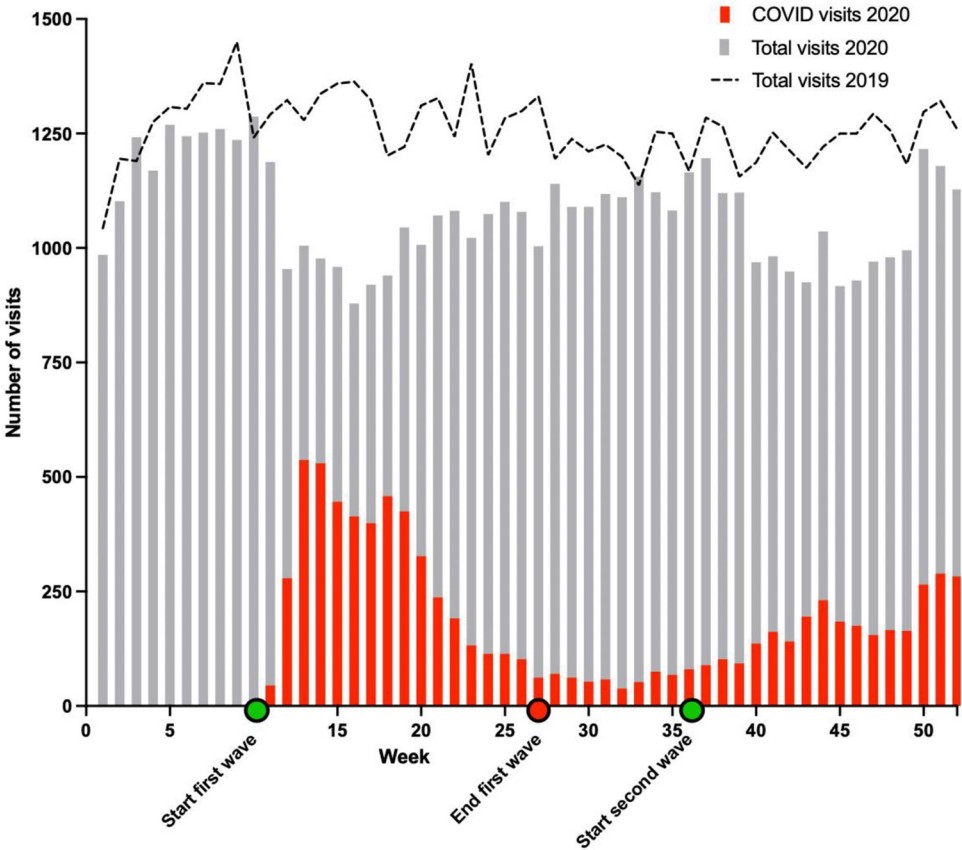

**Fig 1. Overview of patients visiting the ED over 2019 and 2020.**

## Discussion

During the first two COVID-19 waves in the Netherlands, ED utilization was significantly decreased, with 20,3% during the FW and 15.3% during the SW. When comparing the two waves with their reference periods in 2019, there was an increase in transportation by ambulance, more patients were triaged as high urgent, ED length of stay was prolonged and there were higher ARs, particularly in COVID-19 suspected patients. Furthermore, a decline in trauma-related ED visits was observed. The increased triage urgency, longer ED length of stay and higher ARs may reflect a higher workload of EDs during the pandemic waves in year 2020.

Previous studies corroborate our results and also report a significant drop in ED visits, increase in high urgent visits and a decline in trauma-related visits during both waves [7–9, 11–16]. Declines were observed in nearly all non-COVID conditions, although specific demographic groups, including children and older patients, showed disproportionate declines [17, 18]. However, to our knowledge, this is one of the first multicenter studies comparing ED utilization during the first and second wave as well as differences between COVID-19 and non-COVID related visits. One single-center study from the United Kingdom assessed ED attendances and acute medical admissions during the pandemic (2020–2021) and the prior year (2019). During the first wave of the pandemic, daily ED attendance fell by 37%, medical admissions by 30% and medical bed occupancy by 27%, but all returned to normal within a year. This normalization of ED utilization was not corroborated by our study.

**Table 2. Overview of overall ED utilization during COVID waves and reference periods in 2019.**

| | First wave | Reference period first wave | Second wave | Reference period second wave | p value first wave vs reference | p value second wave vs reference | p value first vs second wave |
|---|---|---|---|---|---|---|---|
| Total number of patients, n (%) | 18,024 (14.7%) | 22,607 (18.4%) | 18,282 (14.9%) | 21,557 (17.6%) | <0.001 | <0.001 | 0.608 |
| Age in years, median (IQR) | 62 (39–76) | 59 (32–75) | 62 (37–76) | 60 (33–76) | <0.001 | <0.001 | 0.011 |
| Sex (missing = 23) | | | | | | | |
| Female, n (%) | 8,722 (48.4%) | 11,242 (49.7%) | 8,867 (48.5%) | 10,697 (49.7%) | <0.001 | <0.001 | 0.754 |
| Male, n (%) | 9,298 (51.6%) | 11,363 (50.3%) | 9,415 (51.5%) | 10,856 (50.3%) | <0.001 | <0.001 | 0.866 |
| Possible COVID, n (%) | 4,777 (26.5%) | 0 (0%) | 3,102 (17.0%) | 0 (0%) | - | - | <0.001 |
| Transportation to ED: Ambulance | | | | | | | |
| Self-referral, n (%) | 3,801 (21.1%) | 3,716 (16.7%) | 4,053 (22.2%) | 3,745 (17.8%) | 0.811 | <0.007 | <0.044 |
| GP, n (%) | 2,924 (16.3%) | 3,110 (14.0%) | 2,744 (15.0%) | 3,035 (14.4%) | 0.125 | <0.002 | 0.126 |
| Transportation to ED: Own transport | | | | | | | |
| Self-referral, n (%) | 877 (4.9%) | 1,402 (6.3%) | 966 (5.3%) | 1,287 (6.1%) | <0.001 | <0.001 | 0.231 |
| GP, n (%) | 8,324 (46.2%) | 11,192 (50.4%) | 8,079 (44.2%) | 10,353 (49.1%) | <0.001 | <0.001 | 0.301 |
| Other, n (%) | 2,081 (11.6%) | 4,222 (19.0%) | 2,432 (13.3%) | 2,675 (12.7%) | <0.001 | <0.009 | <0.001 |
| Urgency of ED visit* | | | | | | | |
| high urgency, n (%) | 3,823 (22.9%) | 4,247 (19.8%) | 3,817 (22.6%) | 4,169 (20.5%) | <0.001 | <0.001 | >0.999 |
| low urgency, n (%) | 12,847 (77.1%) | 17,125 (80.2%) | 13,043 (77.4%) | 16,182 (79.5%) | <0.001 | < 0.001 | 0.686 |
| Trauma, n (%) (missing = 1) | 4,933 (27.4%) | 7,375 (32.6%) | 5,251 (28.7%) | 6,916 (32.1%) | <0.001 | <0.001 | 0.004 |
| Median minutes spent in ED, m (IQR) | 157 (105–219) | 147 (94–208) | 154 (99–215) | 147 (96–207) | <0.001 | <0.001 | <0.001 |
| Admission after ED visit, n (%) (missing = 15650) | 8,990 (57.3%) | 10,490 (52.3%) | 8,674 (56.7%) | 10,146 (46.3%) | <0.001 | <0.001 | 0.218 |

if the total number of patients does not match the total of inclusions, this is due to missing value

* using the MTS, red or orange triage colour are classified as high urgency (seen by doctor <10 minutes) and triage colour yellow, green or blue are classified as low urgency (1–3 hours before assessment)

** After Bonferroni correction a p-value of 0.003 was considered statistically significant.

IQR = interquartile range

The reduced ED utilization during the two pandemic waves could be attributed to several possible causes, including pure lockdown effects, fear and uncertainties about a novel infectious disease, and misperceptions about the accessibility of EDs [11]. The lockdowns, which included the closure of non-essential shops, offices and the leisure industry, were associated with lower mobility rates and less traffic or workplace accidents [15].

The decline of ED utilizations was more prominent during the FW than during the SW. This may be explained by a reduction of 'viral fear', the increasing knowledge about the virus and repeated encouragements by the government as well as healthcare organizations to seek care when necessary [7]. This observation might also be explained by the rebound effect of

**Table 3. Comparison of ED utilization between possible COVID vs non-COVID ED visits during first and second wave.**

| | First wave (n = 18,024) | | Second wave (n = 18,282) | | Significance | | |
|---|---|---|---|---|---|---|---|
| | COVID | non-COVID | COVID | non-COVID | first wave COVID vs non-COVID | second wave COVID vs non- COVID | first vs second wave COVID |
| Total number of patients (n) | 4,777 (26.2%) | 13,247 (73.8%) | 3,102 (16.9%) | 15,180 (83.1%) | <0.001 | <0.001 | <0.001 |
| Age in years, median (IQR) | 67 (52–77) | 60 (34–75) | 69 (55–79) | 60 (34–75) | <0.001 | <0.001 | 0.508 |
| Sex (missing = 2) | | | | | | | |
| Female, n (%) | 2,236 (46.8%) | 6,471 (48.9%) | 1,425 (45.9%) | 7,435 (49.0%) | <0.001 | <0.001 | <0.001 |
| Male, n (%) | 2,540 (53.2%) | 6,774 (51.1%) | 1,677 (54.1%) | 7,743 (51.0%) | <0.001 | <0.001 | <0.001 |
| Transportation to ED: ambulance (missing n = 24) | | | | | | | |
| Self-referral, n (%) | 990 (20.8%) | 2,841 (21.5%) | 584(14.2%) | 3,520 (85.8%) | <0.001 | <0.001 | <0.001 |
| GP, n (%) | 1,279 (26.8%) | 1,592 (12.0%) | 1,031 (38.5%) | 1,650 (61.2%) | <0.001 | <0.001 | <0.001 |
| Transportation to ED: own transport | | | | | | | |
| Self-referral, n (%) | 142 (3.0%) | 755 (5.7%) | 73 (2.4%) | 885 (5.8%) | <0.001 | <0.001 | <0.001 |
| GP, n (%) | 2,023 (42.4%) | 6,291 (47.5%) | 1,091 (35.2%) | 6,907 (45.6%) | <0.001 | <0.001 | <0.001 |
| Other, n (%) | 336 (7.0%) | 1,756 (13.3%) | 321 (10.4%) | 2,196 (14.5%) | <0.001 | <0.001 | 0.400 |
| Urgency of ED visit* (missing = 1339) | | | | | | | |
| High urgency, n (%) | 1,596 (36.3%) | 2,227 (18.1%) | 1,325 (44.3%) | 2,492 (17.9%) | <0.001 | <0.001 | <0.001 |
| Low urgency, n (%) | 2,799 (63.7%) | 10,048 (81.6%) | 1,664 (55.6%) | 11,379 (81.6%) | <0.001 | <0.001 | <0.001 |
| Median time spent in ED, m (IQR) | 188 (140–243) | 143 (93–207) | 197 (147–253) | 144 (92–204) | <0.001 | <0.001 | <0.002 |
| Admission after ED visit, n (%) (missing = 3072) | 3,085 (74.9%) | 5,760 (50.1%) | 2,117 (79.9%) | 6,466 (51.5%) | <0.001 | <0.002 | 0.777 |

if the total number of patients does not match the total of inclusions, this is due to missing values

*using the MTS, red or orange triage colour are classified as high urgency (seen by doctor <10 minutes) and triage colour yellow, green or blue are classified as low urgency (1–3 hours before assessment)

**After Bonferroni correction a p-value of 0.004 was considered statistically significant.

IQR = interquartile range

delayed healthcare seeking behavior, possibly leading to an exacerbation of the neglected pathological conditions and healthcare damage. Furthermore, the measurements taken by the government were less strict during the SW, which probably resulted in higher mobility rates and increased traffic, explaining the increase of trauma related visits [12]. Finally, during the SW, more elective, non-COVID care was provided. This may have been associated with higher rates of patients who suffered from complications, both postoperatively and medication related.

During the SW, ED utilization still was significantly lower compared to the reference period, possibly resulting from modified community spread of (viral) infections caused by the lockdown, social distancing and other measures during the pandemic [14]. This hypothesis is

supported by the highly unusual respiratory syncytial-virus summer outbreak in July-August 2021, shortly after lockdown restrictions in the Netherlands were lifted [19].

The strengths of this study include its multicenter design, extensive number of patients included, mostly prospective labeling of (non-)COVID-suspection and preclusion of selection bias by including all patients presenting to the three EDs. Limitations include the retrospective nature of this study, prospective labeling may have been different between hospitals, the COVID waves continued past the end of our study period and 9,790 patients (17.3%) have been labeled retrospectively because there was no prospective labeling at the very beginning of the pandemic. Furthermore. we collected limited data on patient characteristics, which did not include comorbidities, performed procedures in the ED (e.g. intubation), ICU admission and survival. Finally, some reported differences, including ED length of stay and mode of transportation to ED, may be significantly different not always clinically significant due to the large sample size, even after Bonferroni correction.

Similar FW and SW dynamics were observed in all three EDs. It is often hypothesized that the reduction of ED visits during the pandemic was predominantly determined by a reduction in non-urgent ED visits [20, 21]. Although this may still be true, this study was performed in the Netherlands, a country with a well-developed primary care system and relatively low numbers of self-referrals. As the observed decline was similar to the declines found in studies performed in other healthcare systems, the role of non-urgent ED visits may be less significant than previously thought. As a result, this study adds to the body of literature about ED utilization during the global COVID-19 pandemic. Future studies should focus on characteristics of the subsequent waves of this ongoing pandemic, including those in the era of covid vaccinations. Furthermore, it would be desirable to know more about why patients delay or avoid emergency care during pandemics.

## Conclusion

During both COVID-19 waves, ED visits were significantly reduced. ED patients were more often triaged as high urgent, the ED length of stay was longer and ARs were increased compared to the reference period in 2019, reflecting a high burden on ED resources. During the FW, the reduction in ED visits was most pronounced. Here, ARs were also higher and patient were more often triaged as high urgency. These findings stress the need to gain better insight into the motives of patients to delay or avoid emergency care during pandemics, as well as to better prepare EDs for future outbreaks.

## Supporting information

**S1 File.**
(DOCX)

**S1 Dataset.**
(SAV)

## Acknowledgments

The study was approved by the medical ethical committee of Zuyderland Medical Center, Heerlen, the Netherlands (METCZ20210031).

## Author Contributions

**Conceptualization:** Robi Dijk, Patricia Plaum, Stan Tummers.

**Data curation:** Robi Dijk, Patricia Plaum.

**Formal analysis:** Robi Dijk, Patricia Plaum.

**Investigation:** Robi Dijk.

**Methodology:** Robi Dijk, Patricia Plaum, Stan Tummers.

**Resources:** Robi Dijk.

**Supervision:** Frits H. M. van Osch, Dennis G. Barten, Gideon H. P. Latten.

**Visualization:** Stan Tummers.

**Writing – original draft:** Robi Dijk, Patricia Plaum.

**Writing – review & editing:** Stan Tummers, Frits H. M. van Osch, Dennis G. Barten, Gideon H. P. Latten.

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
