## [Decision Letter · Decision Letter 0]

8 Aug 2022

PONE-D-22-15700First and Second Wave Dynamics of Emergency Department Utilization during the COVID-19 Pandemic: a retrospective study in 3 hospitals in The NetherlandsPLOS ONE

Dear Dr. van Osch,

Thank you for submitting your manuscript to PLOS ONE. After careful consideration, we feel that it has merit but does not fully meet PLOS ONE’s publication criteria as it currently stands. Therefore, we invite you to submit a revised version of the manuscript that addresses the points raised during the review process.

We look forward to receiving your revised manuscript.

Kind regards,

Yong-Hong Kuo

Academic Editor

PLOS ONE

Journal Requirements:

3. Please include a copy of Table 3 which you refer to in your text on page 8.

Additional Editor Comments:

The manuscript has been reviewed by four referees. Three of them believe the work has potential to be published, while one recommends Reject. All of the referees comment on quite a number of major issues that must be fixed before publication. Please seriously address these concerns. Unsuccessful revision may lead to rejection of the work.

Reviewers' comments:

Reviewer's Responses to Questions

**Comments to the Author**

1. Is the manuscript technically sound, and do the data support the conclusions?

Reviewer #1: Yes

Reviewer #2: Yes

Reviewer #3: Yes

Reviewer #4: Partly

2. Has the statistical analysis been performed appropriately and rigorously? 

Reviewer #1: Yes

Reviewer #2: Yes

Reviewer #3: Yes

Reviewer #4: No

3. Have the authors made all data underlying the findings in their manuscript fully available?

Reviewer #1: No

Reviewer #2: No

Reviewer #3: Yes

Reviewer #4: Yes

4. Is the manuscript presented in an intelligible fashion and written in standard English?

Reviewer #1: Yes

Reviewer #2: Yes

Reviewer #3: Yes

Reviewer #4: Yes

5. Review Comments to the Author

Reviewer #1: Thank the authors very much for this work. Although the COVID-related topic is critically important, I think this work cannot reach the standard of publication.

Major reason:

1. The results and conclusions are not sufficiently innovative. In the past two years, lots of scholars worldwide have already revealed similar trends in ED visits and utilization. This trend is too general.

In this work, the authors studied two specific COVID waves, the conclusions and findings could be more specific and detailed, in order to address some clinical problems, rather than just giving such a general trend.

2. As the authors mentioned in the conclusion, the data closer to the current date would be more attractive and meaningful. The data for 2019 and 2020 are relatively old and could be less guiding.

3. Some statements lack compact logic. Such as Line.198-199, What metric can quantitatively reflect the burden on ED resources?

Minor reason:

The following problems are actually not minor. The authors should be more careful and do the proofreading before submitting.

1. Line.127 & 131, please use a uniform format to cite the figure or table, rather than "Figure 1" and "Fig 1", "table 1" and "Table 1".

2. The tables are significant results, but I can find two "Table 1", and cannot find "Table 3".

Some other problems are also in the figures and tables. This can be summarized that some information in the paragraph was not shown in the figure, but some other important information in the figure and tables was not studied in the paragraph.

I suggest authors can follow their conclusion in this work. Tracking the latest COVID, and making some comparisons among the latest and previous ones.

Reviewer #2: The subject is interesting and topical.

I would advise the author to add data about ICU admission and if it is not possible, to at least discuss it.

I would also ask the author to add to the discussion a comparaison with what has been reported in other EDs/settings.

Reviewer #3: Dear editor,

Thank you for the opportunity to review this manuscript. My recommendation is minor revision:

1- Please mention the weak and strong points of your study

2- The authors do discuss the specifics of the hospital involved in the study regarding comparison of annual admission in the ED with the past COVID-19 period

3- Literature review is so incomplete. Please discuss the following article in the discussion:

Bagi HM, Soleimanpour M, Abdollahi F, Soleimanpour H. Evaluation of clinical outcomes of patients with mild symptoms of coronavirus disease 2019 (COVID-19) discharged from the emergency department. PLoS One. 2021 Oct 21;16(10):e0258697. doi: 10.1371/journal.pone.0258697

4- Discussion should be improved.

5- How did the authors delete the effect of confounding factors from the study outcome?

6- What is the new finding of this study compared to previous ones this field?

7- How did the authors delete the effect of confounding factors from the study outcome?

Reviewer #4: Thanks for the opportunity to review this article written by Osch et al. The authors analysed the data during the FW and SW of COVID-19 lockdowns in three EDs in Netherlands and compared the results with reference points in 2019. The study showed a significant reduction in ED presentations and trauma-related presentations during this time which were shown in other similar studies. They have also reported an increased rate of higher acuity of covid-19 presentations and higher hospital admission.

Although this is an interesting study and reflects on changes in ED presentations during the FW and SW, there are some major issues regarding the statistical reporting and presentation of the results. I suggest the authors revise the manuscript based on the suggestion and re-submit again. In particular, the numbers reported as AR during the FW and SW are not correct and are misrepresented. I have calculated the percentage based on the numbers given in Table 1 and the admission rates between the SW and the reference time point in 2019 were the same at 47%

One of the most critical points is that the authors presented lots of p-values and comparisons in this study. They have to adjust their p-value of 0.05 using Bonferroni’s correction. The study will be elevated if a statistician will be used to assist authors in data analyses, statistical comparisons and data reporting.

The other main issue was the consideration of “chest pain” as one of the symptoms of possible COVID-19 in this study. Chest pain also reported as one of the symptoms, was not a common presentation of possible COVID-19 patients. I wonder if authors are able to see how many presentations were included as “possible COVID presentations” due to chest pain and re-analyse the data?

introduction: paragraph 2:

10,067 died due to COVID-19 during the first wave out of 50,273 infected COVID-19 patients. Would this be a typo? or the mortality rate was around 25% during the first wave (~50K cases / ~ 10K death). This is very high mortality in comparison to other studies reporting between 1 to 10% based on their health care services.

can the authors explain how "patis were able to refuse participation in retrospective studies?" The next paragraph, it is stated that "the ethics committee waived the need for informed consent.” Please explain?

Line 98-101: This is suggested to use the term “re-identifiable data” instead of “pseudonymized”. The last sentence is redundant and can be deleted.

Table1:

Chest pain was not a common COVID-19 presentation during the FW and SW. This was more likely that a patient presented to an ED with a non-covid chest pain rather than covid-19 chest pain. This could potentially skew the data and covid-19 presentation in this study

Line 124: could the author clarify that the results presented in the first line of the result are for the entire 2019 and 2020 or only for the duration of lockdowns in 2020. Please clarify

The first paragraph of the result section is very confusing. As the authors reported ED utilisation during the FW and SW of COID-19, it is highly suggested to only report the data during these two-time points. In addition, it reported 56,719 presentations to the EDs during 2020 (most likely less during the stud period), while 127,060 patients were eligible for inclusion. If the authors included all consecutive presentations to the EDs, why there is a significant discrepancy between ED presentations and eligible patients?

9,034 reported possible covid-19 patients were for the entire 2020 or only the duration of the study (FW and SW?)

Results for FW and SW: should report the actual ED presentations during each lockdown first.

What is the dominator for the calculation of the percentage in FW (26.5%) and SW (17%).

Table 1:

A total of 18,204 presented during the FW, however the exact number of males and females during this period is adding up to 18,020. Please report the missing number or explain the differences between these two sections?

Age: although it is reported that the Age of ED presentations during the FW and SW were statistically different and also in comparison to the 2019 reference point, it is important to note that the difference between the FW and SW is not clinically important.

In addition, it is important that the authors check whether there were any differences between the reference point for FW and SW in 2019. If there is any difference, they have to adjust their data prior to stating there are any statistical differences between FW and SW in 2020. For example, it is better to compare the percentage of changes of the “FW to 2019 reference point” to the “SW to 2019 reference point”. As an example, looking at Table 1, the number of trauma presentations in the reference period first wave is higher than in the second wave reference period. This doesn’t do anything with COVID and could be seasonal changes. The same can be extrapolated from the percentage of admission in 2019 between the two-time points.

The authors stated that “ the admission rate was increased by 5% during the FW and 10% during the SW”. I think this is an oversight when reporting this admission rate (also authors focused on these numbers as a very important finding). ED presentations during the FW were 18,204 and 8,990 patients got admitted (8,990 / 18,204 = 49% and not 57% reported by the authors) and ED presentations during the second wave were 18,282 and 8,674 got admitted (rate of 47%) which is actually the same admission rate in comparison with the 2019 reference point. I urge the authors to check all numbers, percentage and analysis.

Discussion:

It is important to also discuss the higher length of stay in ED for COVID patients in comparison to non-COVID patients

Line 183: what do the authors mean by “This conception is reinforced by a highly unusual respiratory syncytial-virus outbreak in July-August 2021 after loosening of the lockdown measures”?

Figure 1: mark the end of the SW on the X axis

6. PLOS authors have the option to publish the peer review history of their article (what does this mean?). If published, this will include your full peer review and any attached files.

Reviewer #1: No

Reviewer #2: No

Reviewer #3: No

Reviewer #4: **Yes: **A/Prof Hamed Akhlaghi

---

## [Author Response · Author response to Decision Letter 0]

21 Oct 2022

(also included as "Cover letter" in the submission)

Dear Editors,

We would like to thank the reviewers for their valuable and generous comments on the manuscirpt. 

All the comments were taken into account and have been adjusted. 

We believe that the manuscript is now suitable for publication in PlosOne.

On behalf of all the authors,

Frits H.M. van Osch 

 

Adjusted.

Since in the data sharing agreement it was laid out that data shared between the two institutions would not be shared or made available to third parties as it contains confidential information.

.3. Please include a copy of Table 3 which you refer to in your text on page 8.

Now included

 

Reviewer #1: Thank the authors very much for this work. Although the COVID-related topic is critically important, I think this work cannot reach the standard of publication.

Major reason:

1. The results and conclusions are not sufficiently innovative. In the past two years, lots of scholars worldwide have already revealed similar trends in ED visits and utilization. This trend is too general.

In this work, the authors studied two specific COVID waves, the conclusions and findings could be more specific and detailed, in order to address some clinical problems, rather than just giving such a general trend.

We agree that we are not the first to describe this phenomenon. However, to our knowledge this is one of the first multicenter studies that assess the impact of 2 different waves. Furthermore, this study was performed in the Netherlands, a country with a well-developed primary care system and relatively low numbers of self-referrals. Therefore, we believe this study still is a valuable contribution to the literature. We adjusted the conclusion and made our findings more specific in the new manuscript. 

2. As the authors mentioned in the conclusion, the data closer to the current date would be more attractive and meaningful. The data for 2019 and 2020 are relatively old and could be less guiding.

Data for 2019 and 2020 might be relatively old, but this period was the start of the COVID-19 pandemic whit the greatest influence on society wich makes this data the most valuable. Unfortunately, it is not feasible for us to add data for 2021 and 2022 to this analysis.

3. Some statements lack compact logic. Such as Line.198-199, What metric can quantitatively reflect the burden on ED resources?

Burden on ED resources can be reflected by significant increase in high urgent visits (determined by triage following the Manchester Triage System), higher admission rates and longer ED length of stay. 

Minor reason:

The following problems are actually not minor. The authors should be more careful and do the proofreading before submitting.

1. Line.127 & 131, please use a uniform format to cite the figure or table, rather than "Figure 1" and "Fig 1", "table 1" and "Table 1".

Adjusted

2. The tables are significant results, but I can find two "Table 1", and cannot find "Table 3".

Table 3 is now added

Some other problems are also in the figures and tables. This can be summarized that some information in the paragraph was not shown in the figure, but some other important information in the figure and tables was not studied in the paragraph.

I suggest authors can follow their conclusion in this work. Tracking the latest COVID, and making some comparisons among the latest and previous ones.

Thank you for this suggestion. In our opionion, the most relevant findings in the figure and tables are explained in the paragraph. We have deliberately chosen not to describe all data in the tables, only the most relevant. We have added a sentence abouut ED length of stay, which is of course very significant to our daily practice.

 

Reviewer #2: The subject is interesting and topical.

I would advise the author to add data about ICU admission and if it is not possible, to at least discuss it.

I would also ask the author to add to the discussion a comparaison with what has been reported in other EDs/settings

Thank you for this suggestion. As much as we would like to add data from intensive care admissions, this is unfortunately not feasible for this study because we do not have this data. We therefore added the lack of these data to the limitations of the study.

Reviewer #3: 

My recommendation is minor revision:

1- Please mention the weak and strong points of your study

We added our weak and strong points of the study to the discussion

2- The authors do discuss the specifics of the hospital involved in the study regarding comparison of annual admission in the ED with the past COVID-19 period

We would like to apologize, but we do not fully understand this sentence/suggestion.

3- Literature review is so incomplete. Please discuss the following article in the discussion:

Bagi HM, Soleimanpour M, Abdollahi F, Soleimanpour H. Evaluation of clinical outcomes of patients with mild symptoms of coronavirus disease 2019 (COVID-19) discharged from the emergency department. PLoS One. 2021 Oct 21;16(10):e0258697. doi: 10.1371/journal.pone.0258697

Thank you for this reference. Although we enjoyed reading this study, we believe the scope of this study is totally different from our study, and we do not see how we can refer to this study in our discussion.

4- Discussion should be improved.

We have improved our discussion. For example we have added why this study adds body to the already existing literature and why our study is different from other studies. We also did more literature research to improve our discussion. We also we expanded the study's strengths and weaknesses. 

5- How did the authors delete the effect of confounding factors from the study outcome?

The study is of a descriptive nature with multiple comparisons of patient characteristics and ED care parameters between waves, and does not (and cannot) aim to predict an outcome. The ‘outcome’ in this case is in which wave the number originated and whether the exposure sex is equally distributed among the first and second wave. When considering this example, looking at the distribution of sex between the first and second wave, there is no possible confounder that we could have measured, as the distribution of males and females that visit the ED could only be influenced by internal patient factors or factors related to primary care referrals. 

6- What is the new finding of this study compared to previous ones this field?

To our knowledge this is one of the first multicenter studies that assess the impact of 2 different waves. Furthermore, this study was performed in the Netherlands, a country with a well-developed primary care system and relatively low numbers of self-referrals. Therefore, we believe this study still is a valuable contribution to the literature

7- How did the authors delete the effect of confounding factors from the study outcome?

See answer question 5. 

 

Reviewer #4: Thanks for the opportunity to review this article written by Osch et al. The authors analysed the data during the FW and SW of COVID-19 lockdowns in three EDs in Netherlands and compared the results with reference points in 2019. The study showed a significant reduction in ED presentations and trauma-related presentations during this time which were shown in other similar studies. They have also reported an increased rate of higher acuity of covid-19 presentations and higher hospital admission.

Although this is an interesting study and reflects on changes in ED presentations during the FW and SW, there are some major issues regarding the statistical reporting and presentation of the results. I suggest the authors revise the manuscript based on the suggestion and re-submit again. In particular, the numbers reported as AR during the FW and SW are not correct and are misrepresented. I have calculated the percentage based on the numbers given in Table 1 and the admission rates between the SW and the reference time point in 2019 were the same at 47%

One of the most critical points is that the authors presented lots of p-values and comparisons in this study. They have to adjust their p-value of 0.05 using Bonferroni’s correction. The study will be elevated if a statistician will be used to assist authors in data analyses, statistical comparisons and data reporting.

The other main issue was the consideration of “chest pain” as one of the symptoms of possible COVID-19 in this study. Chest pain also reported as one of the symptoms, was not a common presentation of possible COVID-19 patients. I wonder if authors are able to see how many presentations were included as “possible COVID presentations” due to chest pain and re-analyse the data?

Thanks a lot for your suggestions. We adjusted our p-value using the Bonferroni’s correction and adjusted our manuscript. The Bonferroni’s correction is also stated underneath table 2 and table 3. 

We are not able to remove ‘chest pain’ as one of the symptoms of possible covid-19 infections, this is due to the fact that most of the data is labeled prospectively. One of the criteria in that time, to mark a patients as a suspected covid-19 infection, included chest pain. We can not see in our data based on which criteria someone was labeled as a suspected covid-19 infection. 

introduction: paragraph 2:

10,067 died due to COVID-19 during the first wave out of 50,273 infected COVID-19 patients. Would this be a typo? or the mortality rate was around 25% during the first wave (~50K cases / ~ 10K death). This is very high mortality in comparison to other studies reporting between 1 to 10% based on their health care services.

We have checked these numbers and they are correct. We can explain the difference due to our restrictive testing policy during the first because of a lack of testing recourses (with other words: patients who were tested predominantly concerned patients who were critically ill or admitted). We have adjusted the text to explain this situation. 

can the authors explain how "patis were able to refuse participation in retrospective studies?" The next paragraph, it is stated that "the ethics committee waived the need for informed consent.” Please explain?

We explained this suggestion in our manuscript. 

Line 98-101: This is suggested to use the term “re-identifiable data” instead of “pseudonymized”. The last sentence is redundant and can be deleted.

Adjusted

Table1:

Chest pain was not a common COVID-19 presentation during the FW and SW. This was more likely that a patient presented to an ED with a non-covid chest pain rather than covid-19 chest pain. This could potentially skew the data and covid-19 presentation in this study

Please see the comment we made on your suggestion earlier in this rebuttal letter. 

Line 124: could the author clarify that the results presented in the first line of the result are for the entire 2019 and 2020 or only for the duration of lockdowns in 2020. Please clarify

Adjusted

The first paragraph of the result section is very confusing. As the authors reported ED utilisation during the FW and SW of COID-19, it is highly suggested to only report the data during these two-time points. In addition, it reported 56,719 presentations to the EDs during 2020 (most likely less during the stud period), while 127,060 patients were eligible for inclusion. If the authors included all consecutive presentations to the EDs, why there is a significant discrepancy between ED presentations and eligible patients?

9,034 reported possible covid-19 patients were for the entire 2020 or only the duration of the study (FW and SW?)

The total of 127,060 patients eligible for inclusion are the patients visited in 2019 and 2020. In 2019 there where 65,938 visits and in 2020 56,719 visits. We hope this is more clear in the first paragraph in the new manuscript. 

The total of 9034 possible covid-19 patients were for the entire 2020, of of which 4,777 presented during the FW and 3,102 during the SW. That means 1155 possible covid-19 patients presented to the ED in 2020 outside the intended study period (FW + SW)

Results for FW and SW: should report the actual ED presentations during each lockdown first.

 In the new manuscript we implementend this suggestion.

What is the dominator for the calculation of the percentage in FW (26.5%) and SW (17%).

This is from the total amount of patients visiting the ED

Table 1:

A total of 18,204 presented during the FW, however the exact number of males and females during this period is adding up to 18,020. Please report the missing number or explain the differences between these two sections?

This was a mistake in the manuscript and is now adjusted. As stated under table 2 and 3, if the total number of patients does not match the total of inclusions, this is due to missing value . We have added the number of missings in the tables.

Age: although it is reported that the Age of ED presentations during the FW and SW were statistically different and also in comparison to the 2019 reference point, it is important to note that the difference between the FW and SW is not clinically important.

In addition, it is important that the authors check whether there were any differences between the reference point for FW and SW in 2019. If there is any difference, they have to adjust their data prior to stating there are any statistical differences between FW and SW in 2020. For example, it is better to compare the percentage of changes of the “FW to 2019 reference point” to the “SW to 2019 reference point”. As an example, looking at Table 1, the number of trauma presentations in the reference period first wave is higher than in the second wave reference period. This doesn’t do anything with COVID and could be seasonal changes. The same can be extrapolated from the percentage of admission in 2019 between the two-time points.

Thank you for this suggestion, however we believe there are no seasonal changes ecause we compared the study period with the exact same period as a reference. 

The authors stated that “ the admission rate was increased by 5% during the FW and 10% during the SW”. I think this is an oversight when reporting this admission rate (also authors focused on these numbers as a very important finding). ED presentations during the FW were 18,204 and 8,990 patients got admitted (8,990 / 18,204 = 49% and not 57% reported by the authors) and ED presentations during the second wave were 18,282 and 8,674 got admitted (rate of 47%) which is actually the same admission rate in comparison with the 2019 reference point. I urge the authors to check all numbers, percentage and analysis.

We added missing values to table 2 and 3. 

Discussion:

It is important to also discuss the higher length of stay in ED for COVID patients in comparison to non-COVID patients

Length of stay in the ED for COVID-19 patients is added to the discussion

Line 183: what do the authors mean by “This conception is reinforced by a highly unusual respiratory syncytial-virus outbreak in July-August 2021 after loosening of the lockdown measures”?

We agree that this sentence was not very clear. We have rephrased it, so it hopefully makes more sense now.

Figure 1: mark the end of the SW on the X axis

We did not mark the end of the SW on X-axis on purpose because officially the end of 2020 is not the end of the second wave. It is just the end of our study period.

---

## [Decision Letter · Decision Letter 1]

7 Nov 2022

PONE-D-22-15700R1First and Second Wave Dynamics of Emergency Department Utilization during the COVID-19 Pandemic: a retrospective study in 3 hospitals in The NetherlandsPLOS ONE

Dear Dr. van Osch,

Thank you for submitting your manuscript to PLOS ONE. After careful consideration, we feel that it has merit but does not fully meet PLOS ONE’s publication criteria as it currently stands. Therefore, we invite you to submit a revised version of the manuscript that addresses the points raised during the review process.

We look forward to receiving your revised manuscript.

Kind regards,

Yong-Hong Kuo

Academic Editor

PLOS ONE

Journal Requirements:

Additional Editor Comments:

All the referees are satisfied with the revision. However, there are some minor suggestions which the authors shall incorporate into the revision.

Reviewers' comments:

Reviewer's Responses to Questions

**Comments to the Author**

1. If the authors have adequately addressed your comments raised in a previous round of review and you feel that this manuscript is now acceptable for publication, you may indicate that here to bypass the “Comments to the Author” section, enter your conflict of interest statement in the “Confidential to Editor” section, and submit your "Accept" recommendation.

Reviewer #1: (No Response)

Reviewer #2: (No Response)

Reviewer #3: All comments have been addressed

Reviewer #4: All comments have been addressed

2. Is the manuscript technically sound, and do the data support the conclusions?

Reviewer #1: (No Response)

Reviewer #2: Yes

Reviewer #3: Yes

Reviewer #4: Yes

3. Has the statistical analysis been performed appropriately and rigorously? 

Reviewer #1: (No Response)

Reviewer #2: Yes

Reviewer #3: Yes

Reviewer #4: Yes

4. Have the authors made all data underlying the findings in their manuscript fully available?

Reviewer #1: (No Response)

Reviewer #2: Yes

Reviewer #3: Yes

Reviewer #4: Yes

5. Is the manuscript presented in an intelligible fashion and written in standard English?

Reviewer #1: (No Response)

Reviewer #2: Yes

Reviewer #3: Yes

Reviewer #4: Yes

6. Review Comments to the Author

Reviewer #1: Thank the authors for responses.

Further comments:

The current Conclusion part is relatively casual.

Please enhance the Conclusion part, and summarize more details of the characteristics of both FW and SW, respectively.

Reviewer #2: Even if the authors did not provide data according to ICU admission, at least they included it in the limits sections.

Reviewer #3: Thank you so much for your response. The respected authors can discuss and cite the following article in the discussion line 161 (Previous study....,)

Bagi HM, Soleimanpour M, Abdollahi F, Soleimanpour H. Evaluation of clinical outcomes of patients with mild symptoms of coronavirus disease 2019 (COVID-19) discharged from the emergency department. PLoS One. 2021 Oct 21;16(10):e0258697. doi: 10.1371/journal.pone.0258697

Reviewer #4: I have found the authors' responses adequate and can concur that the study has some merits to get published.

I would like to thank the authors for this work and the opportunity to review their article.

7. PLOS authors have the option to publish the peer review history of their article (what does this mean?). If published, this will include your full peer review and any attached files.

Reviewer #1: No

Reviewer #2: No

Reviewer #3: No

Reviewer #4: **Yes: **A/Prof Hamed Akhlaghi

---

## [Author Response · Author response to Decision Letter 1]

25 Nov 2022

See included cover letter:

Reviewer #1: Thank the authors for responses.

Further comments:

The current Conclusion part is relatively casual.

Please enhance the Conclusion part, and summarize more details of the characteristics of both FW and SW, respectively.

Thank you for the suggestion. The conclusion (as well as the abstract) was amended in the new version.

Reviewer #2: Even if the authors did not provide data according to ICU admission, at least they included it in the limits sections.

Reviewer #3: Thank you so much for your response. The respected authors can discuss and cite the following article in the discussion line 161 (Previous study....,)

Bagi HM, Soleimanpour M, Abdollahi F, Soleimanpour H. Evaluation of clinical outcomes of patients with mild symptoms of coronavirus disease 2019 (COVID-19) discharged from the emergency department. PLoS One. 2021 Oct 21;16(10):e0258697. doi: 10.1371/journal.pone.0258697

This reference has been added to the discussion section.

Reviewer #4: I have found the authors' responses adequate and can concur that the study has some merits to get published.

I would like to thank the authors for this work and the opportunity to review their article.

---

## [Decision Letter · Decision Letter 2]

1 Dec 2022

First and Second Wave Dynamics of Emergency Department Utilization during the COVID-19 Pandemic: a retrospective study in 3 hospitals in The Netherlands

PONE-D-22-15700R2

Dear Dr. van Osch,

We’re pleased to inform you that your manuscript has been judged scientifically suitable for publication and will be formally accepted for publication once it meets all outstanding technical requirements.

Kind regards,

Yong-Hong Kuo

Academic Editor

PLOS ONE

Additional Editor Comments (optional):

Based on the referees' recommendations, I recommend Accept.

Reviewers' comments:

Reviewer's Responses to Questions

**Comments to the Author**

1. If the authors have adequately addressed your comments raised in a previous round of review and you feel that this manuscript is now acceptable for publication, you may indicate that here to bypass the “Comments to the Author” section, enter your conflict of interest statement in the “Confidential to Editor” section, and submit your "Accept" recommendation.

Reviewer #1: All comments have been addressed

2. Is the manuscript technically sound, and do the data support the conclusions?

Reviewer #1: (No Response)

3. Has the statistical analysis been performed appropriately and rigorously? 

Reviewer #1: (No Response)

4. Have the authors made all data underlying the findings in their manuscript fully available?

Reviewer #1: (No Response)

5. Is the manuscript presented in an intelligible fashion and written in standard English?

Reviewer #1: (No Response)

6. Review Comments to the Author

Reviewer #1: (No Response)

7. PLOS authors have the option to publish the peer review history of their article (what does this mean?). If published, this will include your full peer review and any attached files.

Reviewer #1: No

---

## [Editor Report · Acceptance letter]

23 Jan 2023

PONE-D-22-15700R2 

First and second wave dynamics of emergency department utilization during the COVID-19 pandemic: a retrospective study in 3 hospitals in The Netherlands 

Dear Dr. van Osch:

I'm pleased to inform you that your manuscript has been deemed suitable for publication in PLOS ONE. Congratulations! Your manuscript is now with our production department. 

Kind regards, 

on behalf of

Dr. Yong-Hong Kuo 

Academic Editor

PLOS ONE